# Super-Hydrophobic Magnetic Fly Ash Coated Polydimethylsiloxane (MFA@PDMS) Sponge as an Absorbent for Rapid and Efficient Oil/Water Separation

**DOI:** 10.3390/polym14183726

**Published:** 2022-09-07

**Authors:** Mengqi Zhao, Xiaoqing Ma, Yuxi Chao, Dejun Chen, Yinnian Liao

**Affiliations:** 1School of Chemical Engineering and Technology, Xinjiang University, Urumqi 830017, China; 2State Key Laboratory of Chemistry and Utilization of Carbon Based Energy Resources, Urumqi 830017, China

**Keywords:** super hydrophobic, magnetic fly ash, polydimethylsiloxane, oil/water separation

## Abstract

In this study, magnetic fly ash was prepared with fly ash and nano-magnetic Fe_3_O_4_, obtained by co-precipitation. Then, a magnetic fly ash/polydimethylsiloxane (MFA@PDMS) sponge was prepared via simple dip-coating PDMS containing ethanol in magnetic fly ash aqueous suspension and solidifying, whereby Fe_3_O_4_ played a vital role in achieving the uniformity of the FA particle coating on the skeletons of the sponge. The presence of the PDMS matrix made the sponge super-hydrophobic with significant lubricating oil absorption capacity; notably, it took only 10 min for the material to adsorb six times its own weight of n-hexane (oil phase). Moreover, the MFA@PDMS sponge demonstrated outstanding recyclability and stability, since no decline in absorption efficiency was observed after more than eight cycles. Furthermore, the stress–strain curves of 20 compression cycles presented good overlap, i.e., the maximum stress was basically unchanged, and the sponge was restored to its original shape, indicating that it had good mechanical properties, elasticity, and fatigue resistance.

## 1. Introduction

With global industrialization, oil spills remain a major source of water pollution and are among the most difficult challenges facing the world today. They are not only harmful to the natural ecosystem, but also have long-term adverse effects on human health and the economy [1,2]. However, treatment methods, such as flotation [3], combustion and linseed oil, have the disadvantages of poor selectivity and low efficiency [4]. 

The contact angle of water droplets on a lotus leaf is about 156°. As such, water droplets can easily slide off the surface of a leaf, taking surface pollutants with them and thereby achieving a self-cleaning effect. This phenomenon known as the “lotus leaf effect”. Inspired by this natural phenomenon, biomimetic superhydrophobic membrane materials based on polymer materials have been developed through polymerization, vapor/liquid phase deposition, membrane transfer, micro-contact printing, electrostatic spinning and other processes. Superhydrophobic materials have a wide range of applications [5,6,7]. In recent years, the surface structures of lotus leaves have been mimicked, and interfaces similar to those of hydrophobic biomaterials with completely different wettability properties for oil and water have been designed and synthesized on the basis of surface chemical bionics. Such structures can effectively achieve efficient separation of oil and water. Superhydrophobic materials prepared by mimicking the surface characteristics of natural superhydrophobic materials are collectively known as biomimetic superhydrophobic materials. With the aim of adsorbing organic pollutants and purifying oily wastewater, research on such structures is rapidly growing in scope [8,9,10].

Superhydrophobicity means that water droplets are spherical on the surface with a contact angle greater than 150 degrees. True intrinsic superhydrophobicity does not exist, and the maximum water contact angle for flat materials is only 119 degrees. However, metals, ceramics and polymers can be made superhydrophobic by certain treatments, i.e., creating a suitable surface roughness or through the modification of a low surface energy material. For example, metals do not have superhydrophobic properties, but if their surface is roughened by corrosion etching and the surface energy is reduced by fluorination, a contact angle of more than 150 degrees can be obtained, thus turning them into superhydrophobic materials. In contrast, the surface energy of polymers is usually very low, making it easier to convert them into superhydrophobic materials [11].

For example, non-stick pans are made of polytetrafluoroethylene, which is superhydrophobic as long as the surface is rough. The lower the surface energy of the material (i.e., the more energy the molecules on the surface of the material have compared to the molecules inside), the better the hydrophobicity. When a low surface energy material has a microscopic rough structure, an air film will be formed between the water droplets and the material, preventing water from wetting the surface and thus forming a superhydrophobic state [12].

The surface energy of synthetic polydimethylsiloxane (PDMS) is only 22 mN/m. The compound is fluorine-free, environmentally-friendly and cheap, and it can be made to be highly soluble in organic solvents with curing agents [13,14]. PDMS has Si-O-Si as its main chain and has both organic groups and inorganic structures. Its Si-O bond length and bond angle are large, and it is easy to deform. As such, it can grafted/coated in the preparation of low surface energy biomimetic materials with special wettability properties, making it possible to achieve selective adsorption or filtration of oil–water mixtures [15,16]. Wang [17] prepared a super-hydrophobic PDMS@Fe_3_O_4_/MS sponge by an impregnation method. The sponge has the advantages of low cost, simple operation and high oil–water separation efficiency. The same author [18] synthesized a hydrophobic PDMS/reduced graphene oxide composite with an adsorption capacity for n-hexane that was 18.5 times its own weight. Zhu [19] connected cement gel particles with PDMS by covalent bonds. This not only changed the morphology of the particles, but also increased their contact angle, providing a research basis for PDMS organic/inorganic hybrid biomimetic materials. Finally, Yu et al. prepared Fe_3_O_4_ nanoparticles by the solvothermal method. They then used ammonium persulfate (APS) as an initiator to synthesize magnetic polystyrene oil-absorbing materials with different coating rates by emulsion polymerization of styrene and divinylbenzene (DVB) at different doses on the nanoparticle surfaces [20].

Fly ash is a kind of solid waste, whose main components are silica and alumina. Fly ash itself has adsorption capacity. Compared with other materials, fly ash has a large specific surface area and a porous structure, and thus, has good adsorption characteristics. The use of fly ash as a raw material has obvious advantages in terms of the required source materials.

The innovation of this study is the preparation of a new composite functional polymeric bionanomaterial, MFA@PDMS, for oily wastewater treatment. Based on the concept of “waste to waste”, the natural porous structure of solid waste fly ash was used to construct a PDMS porous skeleton to make it rough and improve its oil–water separation efficiency [21,22]. Magnetic Fe_3_O_4_ nanoparticles were used to modify the natural porous structure of the fly ash in order to prepare the magnetic porous skeleton. Then, the superhydrophobic bionanomaterials were combined with the magnetic porous skeleton material coating/resin to prepare fluorine-free oil–water separation polymer bionanomaterials.

The preparation process is simple and the resulting magnetic sponges are super hydrophobic and super lipophilic. Additionally, the magnetic properties of the material make it possible to separate oil–water mixtures by an external magnetic field [23]. Magnetic nanomolecular sieve composites have the advantages of high magnetization performance and surface area. In addition, Fe_3_O_4_ nanoparticles were shown to improve the mechanical properties of the sponge, which remained superhydrophobic when it was under tension or compression. The tested sponge also showed good mechanical stability, oil stability and reusability in terms of its superhydrophobicity and oil absorption [21].

## 2. Materials and Methods

### 2.1. Materials

The fly ash (FA) used in this study was obtained from the Kanas Power Plant in Xinjiang Autonomous Region, China. Its chemical composition was as follows: SiO_2_ (45.9 wt%), Al_2_O_3_ (19.03 wt%), CaO (16.39 wt%), SO_3_ (6.01 wt%), Fe_2_O_3_ (5.65 wt%), K_2_O (2.07 wt%), MgO (2.02 wt%) and Na_2_O (0.954 wt%). The PDMS prepolymer and its curing agent were obtained from American Dow Corning. FeCl_2_·4H_2_O, FeCl_3_·6H_2_O and sodium citrate all came from Macklin Biochemical (Shanghai, China) Co., Ltd. Ammonia water was obtained from Sigma-Aldrich (Shanghai, China) Trading Co., Ltd.

### 2.2. Preparation of Magnetic Fly Ash/Polydimethylsiloxane (MFA@PDMS) Sponge

(1) Preparation of magnetic fly ash: First, we poured FeCl_2_·4H_2_O (1.72 g) and FeCl_3_·6H_2_O (4.72 g) into a three-port flask, added 80 mL distilled water, and stirred for 10 min at 80 °C. Then, we added ammonia water dropwise (25 %) until a pH of 10 was reached. Black material could be seen in the process of dropping, and the reaction continued for 30 min after the solution had turned black. Next, 20 mL 0.3 m sodium citrate solution was added and the mixture heated to 90 °C for 30 min. We then added about 8 g of fly ash and continued stirring for 2 h. At the end of the reaction, we applied solid–liquid separation with an external magnetic field and repeated washing with distilled water. Then, we dried the compound at 80 °C for 12 h and, finally, used mechanical grinding to obtain the magnetic fly ash nanomaterials (FA@Fe_3_O_4_).

(2) Preparation of magnetic fly ash @PDMS bionic material: First, we placed 0.20 g magnetic fly ash, 4.00 g PDMS prepolymer and 0.40 g curing agent (10:1) into a beaker. The mixture was stirred, and 2.00 g anhydrous ethanol was added and mixed evenly. We then slowly added 4 mL water, stirred and allowed the mixture to emulsify for 15 min. It was poured into a sand core funnel mold for forming, and then placed in a drying oven at 120 °C for curing for 1.5 h to obtain dark brown spongy magnetic material (MFA@PDMS).

### 2.3. Adsorption Capacity, Separation Efficiency and Reusability Test

(1) To evaluate the adsorption capacity, a certain mass of the material was first immersed in an organic solvent and weighed after reaching mass absorption equilibrium. The measurement was repeated three times and the average value was taken. The adsorption capacity *Q* of the material was calculated according to Formula (1).
(1)$$Q=\frac{\left(M_{1}-M_{0}\right)}{M_{0}}$$
where *Q* (g/g) is the mass-based adsorption capacity, *M*_1_ (g) is the mass of the sponge after oil absorption and *M*_0_ (g) is the mass of the sponge before oil absorption.

(2) Next, we built an oil–water separation device and calculated the oil–water separation efficiency, *R*, of the sponge according to Formula (2).
(2)$$R=\frac{V_{C}}{V_{0}}\times 100\%\;\;$$
where *R* (%) is the oil–water separation efficiency, *V_c_* (mL) is the volume of the original solvent or oil collected after separation and *V*_0_ (mL) is the volume of the original solvent or oil before separation.

(3) The repeatability of the material was initially evaluated using a simple electronic universal testing machine. First, the material was immersed in an organic solvent to achieve maximum absorption and its mass was weighed and recorded as *M*_2_. Afterwards, the absorbed oil was recovered by mechanical compression and the mass of the material was determined again and recorded as *M*_3_. The reusability of the sponge was assessed by absorption-compression cycles, each time weighing the material to calculate the adsorption multiplier (i.e., the weight of the material after the adsorption of an organic solvent/the weight of the material itself).

### 2.4. Wenzel Model

In Equation (3), *r* represents the roughness factor of the solid surface (*r* > 1), which is actually the magnitude of the ratio of the actual area on the surface to the projected area; $\theta_{W}$ represents the apparent contact angle with the rough surface; and $\theta_{Y}$ is the contact angle in an ideal state [12].
(3)$$\frac{\cos\theta_{W}=r\left(\gamma_{SV}-\gamma_{SL}\right)}{\gamma_{LV}}=r\cdot \cos\theta_{Y}$$

Under the condition that the solid surface itself is hydrophilic, if the surface roughness increases, the hydrophobicity of the surface will not increase. In contrast, under the condition that the solid surface itself has a certain degree of hydrophobicity, if the surface roughness increases, it will also have a certain effect on the hydrophobicity of the solid surface, and the level of hydrophobicity will naturally increase.

### 2.5. Determination and Characterization

The composite samples were analyzed using an X-ray fluorescence spectrometer (XRF, ARL PERFORM’X, Thermo Fisher Scientific, Waltham, MA, USA) and Fourier transformation infrared spectrometer (FTIR, VERTEX 70 RAMI, Bruker Corporation, Billerica, MA, USA). The morphologies of samples were comparatively observed by scanning electronic microscopy (SEM, SU8010, Hitachi, Ltd., Tokyo, Japan). The morphologies of the samples were observed using a transmission electron microscope (TEM, JEM-2100, JEOL Ltd. Akishima City, Tokyo, Japan). Magnetic analyses of the samples were conducted using a vibrating sample magnetometer (VSM, Lake Shore Corporation, Columbus, OH, USA). The stress–strain analyses of the samples were conducted using an electronic universal testing machine (CMT6103, MTS Systems Corporation, Eden Prairie, MN, USA).

## 3. Results and Discussion

### 3.1. Materials Characterization

#### 3.1.1. TEM Analysis of Nanometer Fe_3_O_4_

The sample powder was dried using a 200 nm copper mesh. The electron acceleration voltage of the transmission electron microscope (TEM) was 100 kV.

A TEM image of the magnetic nano-Fe_3_O_4_ is shown in Figure 1. As shown, the average particle size of the magnetic nanospheres was 30 nm, and the particle size was uniform.

#### 3.1.2. SEM Analysis

As Figure 2a,b shows, the surface of PDMS was extremely uniform and the folds were extremely smooth, which is the reason why the sponge material had such good mechanical properties and elasticity. Additionally, its internal porous structure increased its oil absorption performance to a certain extent.

As Figure 2c,d shows, the morphology changed significantly with modification with MFA; the smooth surface of the PDMS became rough, which was favorable for hydrophobicity. Additionally, because of the natural porous structure of FA, the MFA@PDMS had a rough, honeycomb porous structure, which undoubtedly increased the hydrophobicity of the sponge surface and the oil absorption ability of the sponge.

#### 3.1.3. FT-IR Analysis

The Fourier transform infrared spectrum (FT-IR) of the MFA@PDMS is shown in Figure 3. The absorption peak at 2964 cm^−1^ corresponds to -CH_3_ stretching vibrations; that at 1262 cm^−1^ was attributed to the sharp symmetric deformation vibration peak of Si–CH_3_; that at 793 cm^−1^ corresponded to that Si-C tensile vibration peak [23]; that near 1083 cm^−1^ was attributed to the asymmetric stretching vibrations of Si-O-Si bond and Si-O-Al bonds, which was caused by the polymerization of the fly ash aluminosilicate or silicon-oxygen tetrahedron structure. Finally, a crystal lattice absorption peak of Fe_3_O_4_ appeared at 599 cm^−1^, indicating that the MFA@PDMS material had been prepared successfully.

#### 3.1.4. Characterization of Magnetic Properties on MFA@PDMS Sponge

The vibrating sample magnetometer (VSM) test results for nano-Fe_3_O_4_ and MFA@PDMS sponge are shown in Figure 4. As shown, their coercivity was low, and they demonstrated super-paramagnetism. Figure 5 is a graph comparing the natural sedimentation separation and magnetic field separation of the FA@Fe_3_O_4_ suspension.

### 3.2. Hydrophobicity of MFA@PDMS Sponge

Deionized water dyed with blue ink (aqueous phase) and hexane dyed with oil-soluble black (oil phase) were dropped on the surface of the MFA@PDMS sponge (the volume ratio of hexane to water was 1:1). The experimental results are shown in Figure 6. The deionized water with blue ink formed a typical spherical shape, revealing the superhydrophobicity of the surface of the material. The surface contact angle was calculated to be greater than 150°, while the black hexane penetrated the interior of the parent body. A material with hydrophobic characteristics cannot be wetted by water, but can be wetted by oil. It can be seen that the sponge material had good hydrophobicity to aqueous solution, reflecting its good waterproof performance; this is also key its selectivity for oil–water separation.

### 3.3. Oil–Water Separation Performance of MFA@PDMS Sponge

#### 3.3.1. Separation Performance for Immiscible Oil/Water

In order to verify the oil–water separation capability of the MFA@PDMS sponge, an experiment was designed (see Figure 7). The MFA@PDMS sponge was placed in a normal hexane aqueous solution containing Sudan red II dye. The hexane solution was then moved to an oily area using a magnet. It was observed that the red normal hexane was quickly absorbed by the sponge (the red circles correspond to the locations of the red hexane), with the oil-bearing water quickly becoming transparent, indicating that that material has excellent oil–water separation capability (the separation efficiency can reach more than 95%), can effectively and quickly remove n-hexane (oil phase) from water, and can be remotely controlled via an external magnetic field, thereby facilitating the separation, recovery and transportation of oils.

According to the experimental process shown in Figure 8, the adsorption times of pure PDMS, FA@PDMS, Fe_3_O_4_@PDMS and MFA@PDMS sponges for n-hexane were tested; the results are shown in Figure 8. As indicated, the adsorption multiple of the MFA@PDMS sponge was the highest, reaching six times of its own weight, indicating that the addition of MFA significantly improved the oil–water separation effect of the PDMS.

#### 3.3.2. Separation Performance for Oil–Water Emulsion

In general, the particle size of the dispersed phase of oil–water emulsions is much smaller than 20 μm, and a stable protective layer is formed on the surface of droplet molecules. As such, it is difficult to break emulsions, making oil–water separations extremely difficult. In this study, the separation efficiency of MFA@PDMS for different types of emulsified oils (n-hexane, phenol and canola oil) was investigated via the experimental method shown in Figure 9. The results are shown in Figure 10.

This oil–water separation device was based on the hydrophobicity and lipophilicity of the MFA@PDMS based material. The MFA@PDMS was clamped in the middle of the left tube and the oil–water mixture was poured into the wide-mouth collection container on the right side. The red oil quickly passed through the MFA@PDMS and flowed into the collection container on the lower left side, while the blue deionized water was blocked on the top side of the MFA@PDMS. Finally, the oil–water mixture was successfully separated. When the oil–water mixture came into contact with the MFA@PDMS surface, its superhydrophobic and super lipophilic properties caused the oil contaminants to penetrate and pass through the material instantly, while the aqueous solution was blocked by its hydrophobic nature. In this study, after each separation, the sample was washed and dried before being reused for the next oil–water separation.

As shown in Figure 10, the separation efficiency of magnetic MFA@PDMS sponge for n-hexane emulsified oil was the highest, reaching 97.12%, followed by phenol, which reached 95.27%. This was due to the unique structure of fly ash, i.e., its large specific surface area, porosity and abundant aluminum-silicon components, as well as the super-hydrophobic performance of PDMS, which can attract organic molecules and chain-like structures to the inner surface of sponge materials through π-π dispersion interactions and the donor receptor effect [6].

### 3.4. Reuse of MFA@PDMS Sponge

The experiment shown in Figure 11 was designed to characterize the reutilization performance of the sponge bionic material. The sponge material was put into n-hexane dyed with Sudan red II, and it was observed that its thickness increased from 0.6 cm to 1.2 cm after only 10 min. We then removed the sponge bionic material and weighed it to calculate its adsorption multiple (i.e., the weight of the material after absorbing the normal hexane/the weight of the material). The oil phase could be extruded by simple compression, and the material could be reused within 30 min after extrusion. Over eight adsorption-extrusion-drying tests, the oil absorption capacity did not decrease significantly, showing good reusability.

An H5KT-type static mechanical tester was used for our stress–strain performance analysis. The compressive stress–strain relationships under single and 20 cyclic quasi-static compressive loadings are shown in Figure 12 and Figure 13, respectively (the compression rate was 50 mm/min).

As shown in Figure 12 and Figure 13, the maximum stress of the material was 446.9 kPa before adsorption on the sponge and 410 kPa afterwards, indicating that the material has good recyclability. The stress–strain curves of 20 compression cycles showed good overlap, and the maximum stress was basically unchanged, indicating that the sponge has good mechanical properties, elasticity, and fatigue resistance.

## 4. Conclusions

In this study, magnetic FA was used to construct a flexible PDMS porous structure, and a MFA@PDMS sponge for oil–water separation was successfully prepared. The sponge was shown to be able to adsorb six times its own weight of n-hexane in only 10 min. It could remotely process oil-bearing waters under the action of an external magnetic field, making it an effective, fast and convenient approach. The MFA@PDMS can be reused by applying a simple oil absorption-extrusion-drying operation which takes around 30 min, after which the oil–water separation performance is not significantly reduced. The stress–strain curves of 20 compression cycles for the sponge showed good overlap, and the maximum stress was basically unchanged, i.e., the sponge was restored to its original shape, indicating that it has good mechanical properties, elasticity, and fatigue resistance.

## Figures and Tables

**Figure 1 polymers-14-03726-f001:**
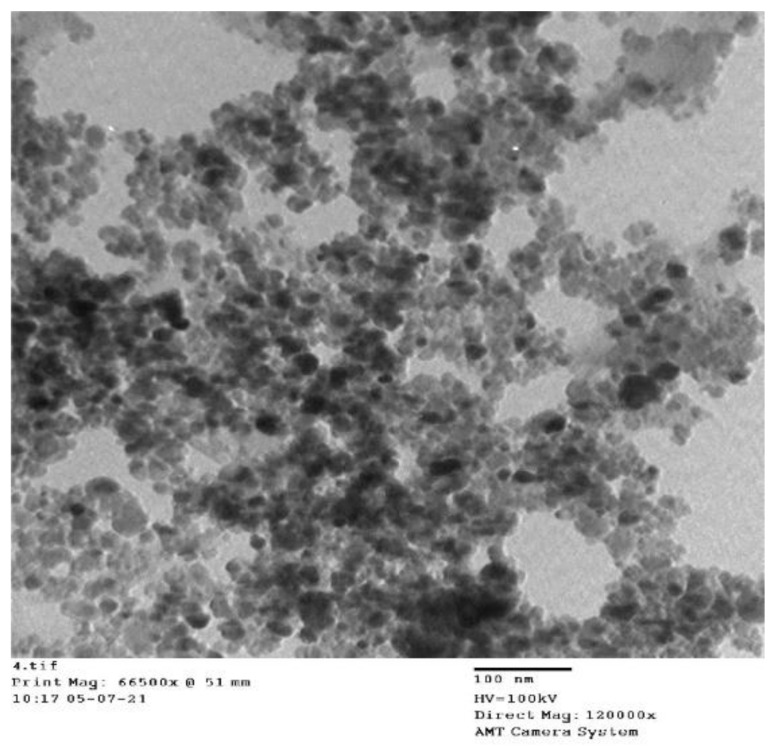
Transmission Electron Microscope Image of Fe_3_O_4_.

**Figure 2 polymers-14-03726-f002:**
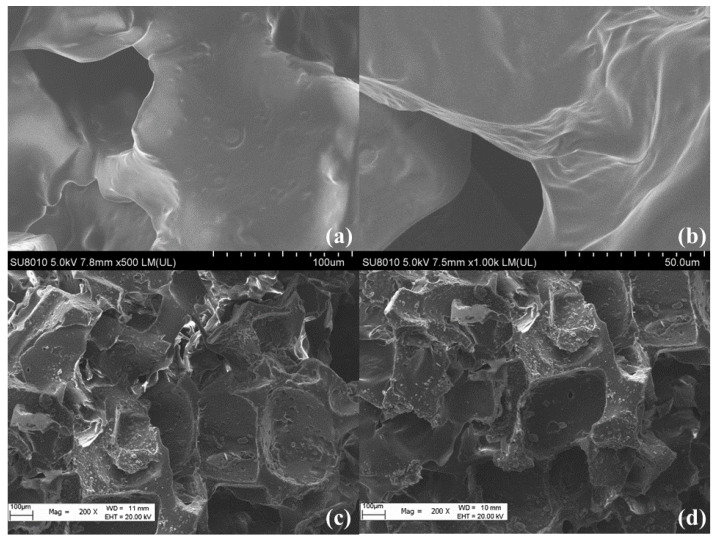
SEM of (**a**,**b**) PDMS and (**c**,**d**) MFA@PDMS.

**Figure 3 polymers-14-03726-f003:**
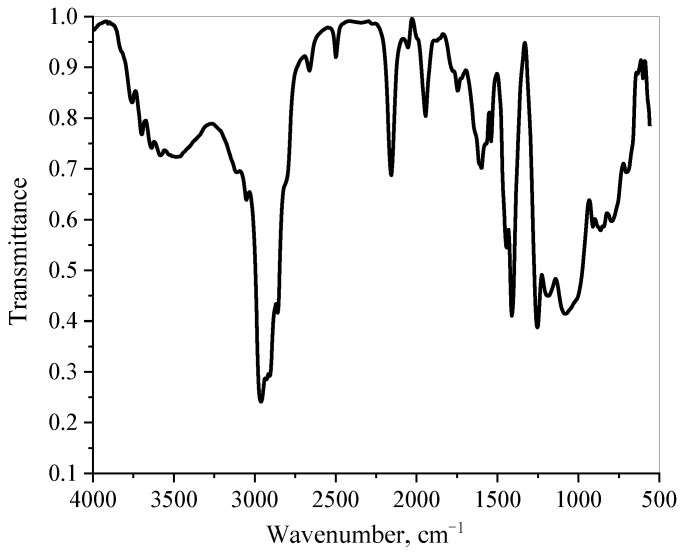
FT-IR Diagram of MFA@PDMS.

**Figure 4 polymers-14-03726-f004:**
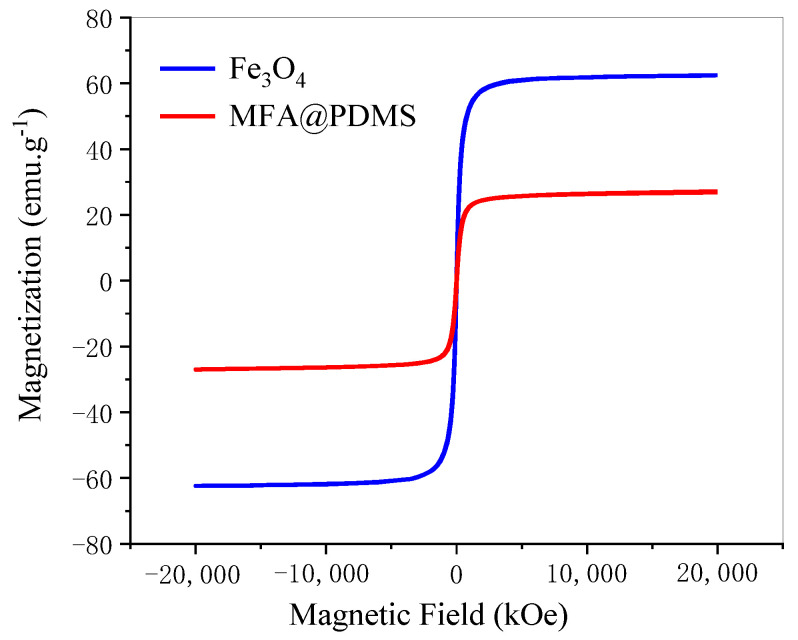
Hysteresis loop of Fe_3_O_4_ and MFA@PDMS.

**Figure 5 polymers-14-03726-f005:**
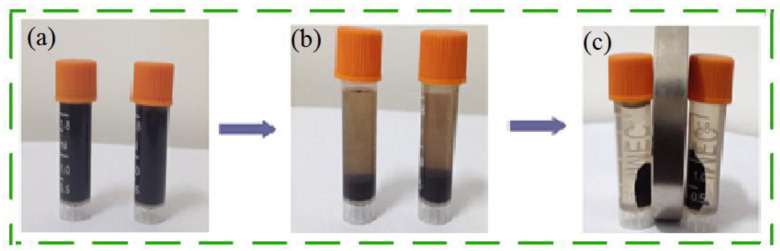
Solid–liquid separation process of FA@Fe_3_O_4_. (**a**) Suspension; (**b**) Natural sedimentation separation; (**c**) Magnetic field separation.

**Figure 6 polymers-14-03726-f006:**
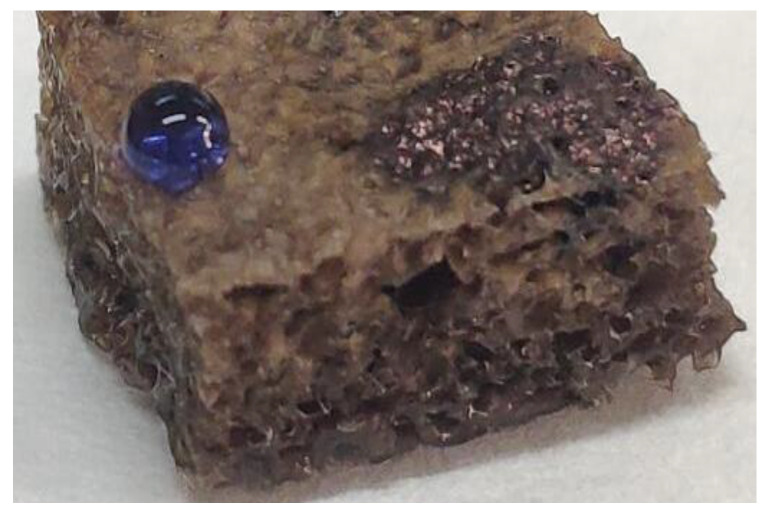
Hydrophobic Properties of MFA@PDMS sponge.

**Figure 7 polymers-14-03726-f007:**
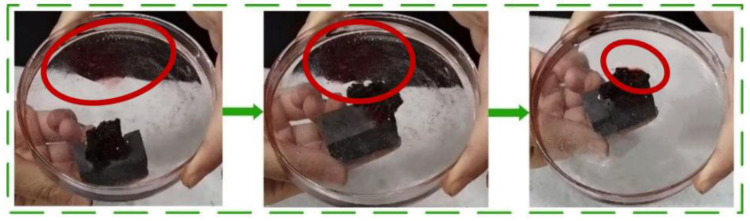
Oil–water separation using the MFA @ PDMS sponge.

**Figure 8 polymers-14-03726-f008:**
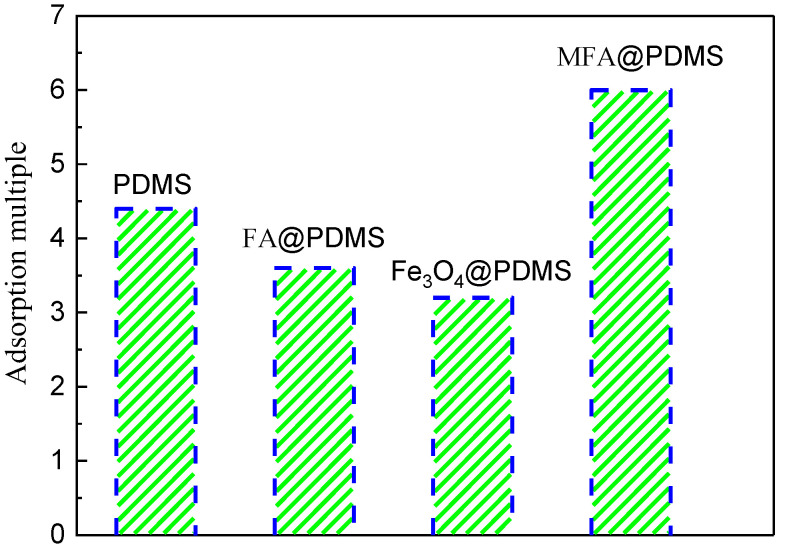
Comparison of adsorption times.

**Figure 9 polymers-14-03726-f009:**
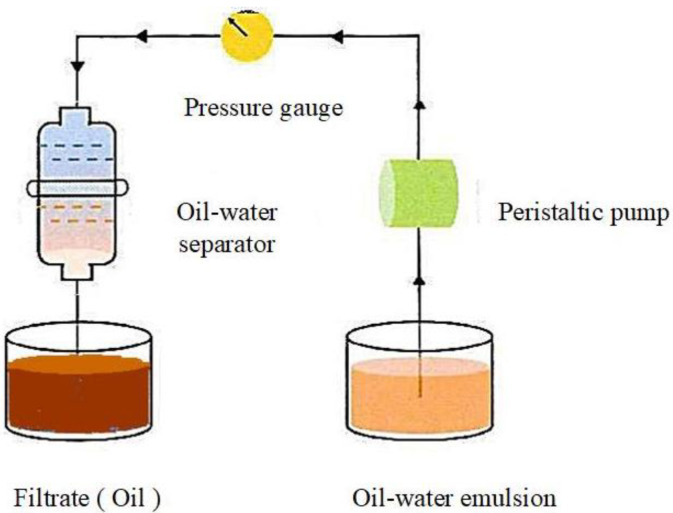
Schematic diagram of oil–water separation process of emulsion mixture.

**Figure 10 polymers-14-03726-f010:**
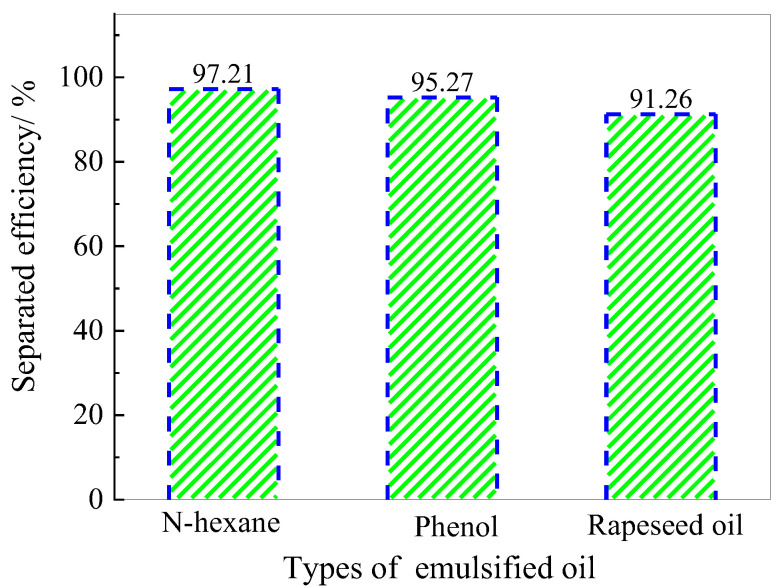
Schematic diagram of emulsion oil–water separation efficiency changes over time.

**Figure 11 polymers-14-03726-f011:**
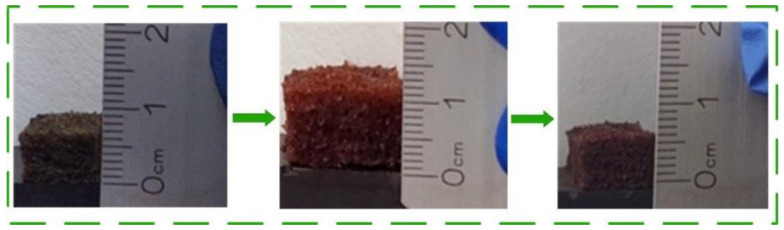
Flexible deformation of materials during oil absorption-extrusion-drying process on MFA@PDMS sponge.

**Figure 12 polymers-14-03726-f012:**
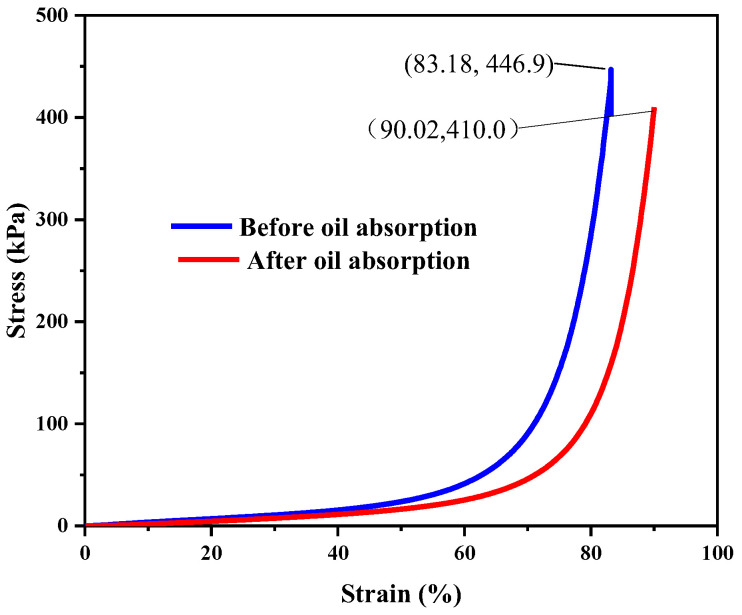
Stress–strain diagram of MFA@PDMS sponge material during a single cycle.

**Figure 13 polymers-14-03726-f013:**
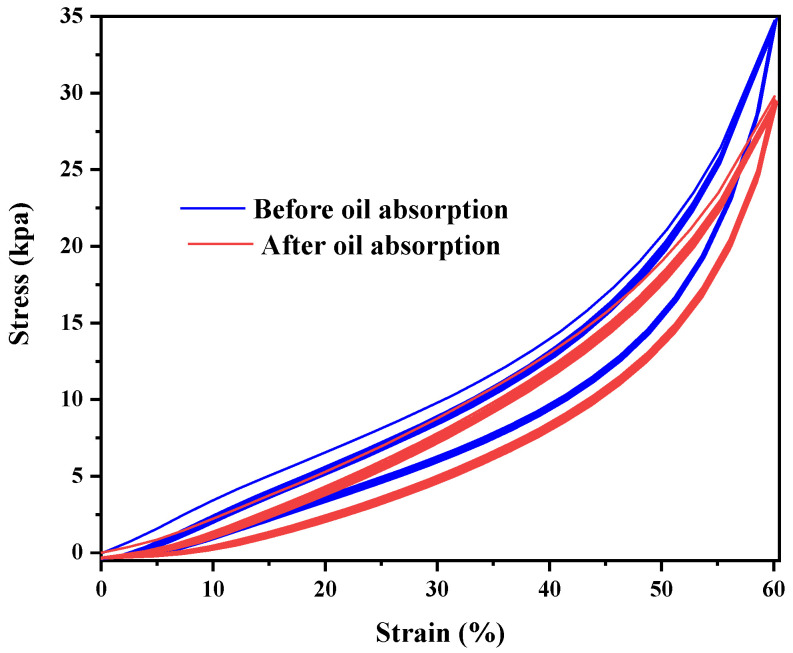
Stress–strain diagram of MFA@PDMS sponge over 20 cycles.

## Data Availability

The data presented in this study are available on request from the corresponding author.

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
