# Peer review of "Super-Hydrophobic Magnetic Fly Ash Coated Polydimethylsiloxane (MFA@PDMS) Sponge as an Absorbent for Rapid and Efficient Oil/Water Separation"

_polymers, 2022, doi:10.3390/polym14183726_

Round 1

Reviewer 1 Report (Previous Reviewer 1)

The authors have made some corrections, but some problems still need to be revised.

General comments: 

1. Line 100-102, the author mentioned “ and the magnetic sponge not only has super hydrophobicity and super lipophilicity, but also can achieve the purpose of separating oil-water mixture under the manipulation of the applied magnetic field”, the separating oil-water mixture performance is mainly due to the sponge or the MFA should be explained clearly.

2. TEM analysis of MFA@PDMS and Scanning Electron Microscopy analysis of FA, PDMS, Fe3O4 and MFA@PDMS are advised to be provided to make a further morphology comparison of those materials.

3. The FT-IR analysis of FA, PDMS and Fe3O4 are advised to be provided to compare the absorption peaks and to demonstrate that the absorption peaks are caused by the polymerization of fly ash aluminosilicate or silicon-oxygen tetrahedron structure.

4. Figure 5, the conclusion “it can be observed that the red normal hexane is quickly absorbed by the sponge, the oil-bearing wat body quickly recovers to be clear and transparent,” something with red color still can be seen and the difference before and after separation is very small in those three pictures. More clear pictures should be provided, or explanations should be provided.

Author Response

Comments from reviewer 1

The authors have made some corrections, but some problems still need to be revised.

Dear reviewer, thank you very much for your patience in reviewing this paper, and we apologize for any omissions. However, we are glad that we have corrected and added to your suggestions and shortcomings. Thank you again for your review!

  1. Line 100-102, the author mentioned “ and the magnetic sponge not only has super hydrophobicity and super lipophilicity, but also can achieve the purpose of separating oil-water mixture under the manipulation of the applied magnetic field”, the separating oil-water mixture performance is mainly due to the sponge or the MFA should be explained clearly.

Dear Reviewer, thank you for your kind attention to pointing out our mistakes, and we have made corrections to the problems for your review.

The preparation process of magnetic materials is simple and magnetic sponges are not only super hydrophobic and super lipophilic, but also can achieve the purpose of separating oil-water mixtures, and the magnetic properties of the material allow this process to be manipulated by an external magnetic field [17]. Moreover, the magnetic nanomolecular sieve composites have the advantages of high magnetization performance and surface area, and in addition, Fe3O4 nanoparticles can also improve the mechanical properties of the sponge, which remains superhydrophobic when it is strained in tension or compression, and the tested sponge also shows good mechanical stability, oil stability and reusability in terms of superhydrophobicity and oil absorption [15].

  1. TEM analysis of MFA@PDMS and Scanning Electron Microscopy analysis of FA, PDMS, Fe3O4 and MFA@PDMS are advised to be provided to make a further morphology comparison of those materials.

Dear reviewer, we sincerely thank you for your valuable comments and suggestions, and we have done our best to add the missing relevant characterization materials, please take a look.

Figure 2. SEM of (a), (b) PDMS and (c), (d) MFA@PDMS.

As Figure 2 (a, b) shows the SEM images of PDMS, it can be found that the surface of PDMS is extremely uniform and the folds are also extremely smooth, which is the reason why the sponge material prepared by PDMS has good mechanical properties and elasticity. It also has a porous structure inside, which increases the oil absorption performance to a certain extent.

As Figure 2 (c, d) shows the SEM images of MFA@PDMS, the morphology changed significantly under the compound modification with MFA, the original smooth surface of PDMS became rough, and the increase of roughness is favorable for the hydrophobicity of sponge material. Also because of the natural porous structure of fly ash, MFA@PDMS has a rough structure and at the same time a honeycomb porous structure, which is undoubtedly beneficial to the hydrophobicity of the sponge surface and the oil absorption ability of the sponge.

  1. The FT-IR analysis of FA, PDMS and Fe3O4 are advised to be provided to compare the absorption peaks and to demonstrate that the absorption peaks are caused by the polymerization of fly ash aluminosilicate or silicon-oxygen tetrahedron structure.

Thank you very much for your meticulous suggestions and comments. In the IR spectrum, asymmetric stretching vibrations of Si-O-Si (Al) have appeared, which is a sign of the polymerization of silica-oxygen tetrahedra in fly ash.

The absorption peak near 1083 cm-1 is ascribe to that asymmetric stretching vibration of Si-O-Si bond and Si-O-Al bond, which is cause by the polymerization of fly ash aluminosilicate or silicon-oxygen tetrahedron structure.

  1. Figure 5, the conclusion “it can be observed that the red normal hexane is quickly absorbed by the sponge, the oil-bearing wat body quickly recovers to be clear and transparent,” something with red color still can be seen and the difference before and after separation is very small in those three pictures. More clear pictures should be provided, or explanations should be provided.

Thank you very much for your valuable suggestions, we have already replaced the pictures, thank you again!

In order to verify the oil-water separation capability of the MFA@PDMS sponge, an experiment as shown in Figure 6 is designed, the MFA@PDMS sponge is placed in a normal hexane aqueous solution containing Sudan red II dyeing, the normal hexane aqueous solution is moved to an oil area under the drive of a magnet, and it can be observed that the red normal hexane is quickly absorbed by the sponge (the location of the red circles is the location of the red hexane), the oil-bearing wat body quickly recovers to be clear and transparent, which indicate that that material has excellent oil-water separation capability (the separation efficiency can reach more than 95 %), can effectively and quickly remove n-hexane (oil phase) in the water, and can be remotely controlled through an external magnetic field, thereby greatly facilitating the separation, recovery and transportation of the oil stain.

Figure 6. Oil-water separation performance on MFA @ PDMS sponge.

  Finally, we sincerely hope that this revised manuscript has addressed all your comments and suggestions. We appreciated for reviewer’s warm work earnestly, and hope that the correction will meet with approval. Once again, thank you very much for your comments and suggestions. We would like to thank the referee again for taking the time to review our manuscript.

Reviewer 2 Report (New Reviewer)

This study focuses on coating superhydrophobic fly ash onto a polydimethylsiloxane sponge for fast and efficient oil/water separation via adsorption technique. The application area is very interesting along with the idea. However, the manuscript's writing quality is poor and there are some serious issues such as incorrect figure calling and missing figures. the experimental procedures are also not clearly stated. it also lacks in providing error analysis or repeatability testing. Therefore, I would recommend reconsidering the manuscript after the authors can address all the below-mentioned issues.        

1. The introduction section of the manuscript is somewhat monotonous. It provides a lot of background information; however, the novelty of this study is not clearly mentioned. 

2. The process used by the authors to measure the adsorption capacity of the material can be erroneous as it will also include the solvent/oil on the surface of the material due to viscosity. This sticking oil or solvent will also be present in the measured weight which can be random as well. Therefore, the authors must provide repeatability test results. 

3. Lines 164-168 should be rewritten for clarity. 

4. The TEM image provided in Fig. 1 is of poor quality and it is hard for the readers to follow the provided information from the figure. Authors can provide another image with high resolution. 

5. Line 189 the call for Figure 1 is incorrect. 

6. Line 202, there is mention of Figure 4 containing a graph. however, in the manuscript, Figure 4 is something else. 

7. Could the authors provide details of Figure 7 Experimental Procedure (Line 250)

8. The reusability of the Modified sponge is only shown for hexane. What about other oils? and the exact amount of decrement of adsorption capacity is not mentioned after eight cycles. 

Author Response

Comments from reviewer 2

This study focuses on coating superhydrophobic fly ash onto a polydimethylsiloxane sponge for fast and efficient oil/water separation via adsorption technique. The application area is very interesting along with the idea. However, the manuscript's writing quality is poor and there are some serious issues such as incorrect figure calling and missing figures. the experimental procedures are also not clearly stated. it also lacks in providing error analysis or repeatability testing. Therefore, I would recommend reconsidering the manuscript after the authors can address all the below-mentioned issues. 

Dear reviewer, thank you very much for your patience in reviewing this paper, and we apologize for any omissions. However, we are glad that we have corrected and added to your suggestions and our own shortcomings, such as the problems with the pictures and the missing experimental procedures. Thank you again for your review!

  1. The introduction section of the manuscript is somewhat monotonous. It provides a lot of background information; however, the novelty of this study is not clearly mentioned. 

Dear reviewer, we have rewritten the introduction to highlight the innovative points of this study for the problems you said you pointed out.

The innovation of this study is to prepare a new composite functional polymeric bionanomaterial MFA@PDMS for oily wastewater treatment. Based on the concept of "waste to waste", the natural porous structure of solid waste fly ash is used to construct PDMS porous skeleton to make it rough and improve the oil-water separation efficiency [15-16]. And the magnetic Fe3O4 nanoparticles were used to modify the natural porous structure of solid waste fly ash to prepare the magnetic porous skeleton, and then the superhydrophobic bionanomaterials were combined with the magnetic porous skeleton material coating/resin to successfully prepare the fluorine-free oil-water separation and green functional polymer bionanomaterials.

  1. The process used by the authors to measure the adsorption capacity of the material can be erroneous as it will also include the solvent/oil on the surface of the material due to viscosity. This sticking oil or solvent will also be present in the measured weight which can be random as well. Therefore, the authors must provide repeatability test results. 

Dear reviewer, thank you very much for your query, which we deeply understand. There are two main representations for testing the oil absorption properties of materials, one is expressed in terms of volume expansion and the other in terms of mass ratio. In this study, the mass ratio representation was used. At the same time, during the experiments, we are carrying out the procedure of repeated measurements to take the average value. The specific information is as follows.

(1) To evaluate the adsorption capacity of the material, a certain mass of the material was first immersed in the organic solvent and weighed after reaching mass absorption equilibrium. The measurement was repeated three times and the average value was taken. The adsorption capacity Q of the material is calculated according to the following formula (1).

Where: Q (g/g) is the mass-based adsorption capacity, M1 (g) is the mass of the sponge after oil absorption, and M0 (g) is the mass of the sponge before oil absorption.

  1. Lines 164-168 should be rewritten for clarity. 

Thank you for your criticism of the shortcomings of this article, we have made changes, thank you very much!

2.3. Wenzel model

(3)

In equation 3, r represents the roughness factor in the solid surface (r > 1), which is actually the magnitude of the ratio of the actual area on the surface to the projected area;  represents the apparent contact angle with the rough surface; and  is the contact angle in the ideal state [23].

It can be seen through the formula: Under the condition that the solid surface itself is hydrophilic, if the surface roughness is made to increase, the hydrophobicity of the surface will not increase; Under the condition that the solid surface itself has a certain degree of hydrophobicity, if the surface roughness is made to increase, it will also have a certain effect on the hydrophobicity of the solid surface, and the hydrophobicity will naturally increase.

  1. The TEM image provided in Fig. 1 is of poor quality and it is hard for the readers to follow the provided information from the figure. Authors can provide another image with high resolution. 

Dear reviewer, we sincerely thank you for your valuable comments and suggestions, and we have added some more SEM images of the material for your review.

Figure 2. SEM of (a), (b) PDMS and (c), (d) MFA@PDMS.

As Figure 2 (a, b) shows the SEM images of PDMS, it can be found that the surface of PDMS is extremely uniform and the folds are also extremely smooth, which is the reason why the sponge material prepared by PDMS has good mechanical properties and elasticity. It also has a porous structure inside, which increases the oil absorption performance to a certain extent.

As Figure 2 (c, d) shows the SEM images of MFA@PDMS, the morphology changed significantly under the compound modification with MFA, the original smooth surface of PDMS became rough, and the increase of roughness is favorable for the hydrophobicity of sponge material. Also because of the natural porous structure of fly ash, MFA@PDMS has a rough structure and at the same time a honeycomb porous structure, which is undoubtedly beneficial to the hydrophobicity of the sponge surface and the oil absorption ability of the sponge.

  1. Line 189 the call for Figure 1 is incorrect. 

Thank you very much for your careful review, we have revised it, thank you again!

The Fourier transform infrared spectrum (FT-IR) of the MFA @PDMS is shown in Figure 3. As seen from Figure 3, that absorption peak at 2964 cm-1 correspond to -CH3 stretching vibration; 1262 cm-1 attributed to the sharp symmetric deformation vibration peak of Si − CH3; The absorption peak at 793 cm-1 correspond to that Si-C tensile vibration peak [17]; The absorption peak near 1083 cm-1 is ascribe to that asymmetric stretching vibration of Si-O-Si bond and Si-O-Al bond, which is cause by the polymerization of fly ash aluminosilicate or silicon-oxygen tetrahedron structure; The crystal lattice absorption peak of Fe3O4 appeared at 599 cm-1, which indicated that the MFA@PDMS material had been prepared successfully.

  1. Line 202, there is mention of Figure 4 containing a graph. however, in the manuscript, Figure 4 is something else. 

Dear reviewer, we sincerely thank you for your kind reminders, we have added the missing images and we are very sorry for our negligence.

Figure 5. Solid-liquid separation process of CFA@Fe3O4. (a) Suspension (b) Natural sedimentation separation (c) Magnetic field separation.

  1. Could the authors provide details of Figure 7 Experimental Procedure (Line 250)

Dear reviewer, we have added and clarified your suggestions, and we thank you very much for your corrections.

Figure 9. Schematic diagram of oil-water separation process of emulsion mixture.

In general, the particle size of the dispersed phase of oil-water emulsions is much smaller than 20 μm, and a stable protective layer is formed on the surface of the droplet molecules, so it is difficult to break the emulsions, making the oil-water separation extremely difficult. In this study, the separation efficiency of MFA@PDMS for different types of emulsified oils (n-hexane, phenol and canola oil) was investigated according to the experimental method in Figure 9, and the results are shown in Figure 10.

This oil-water separation device is based on the hydrophobicity and lipophilicity of the MFA@PDMS based material. The MFA@PDMS is clamped in the middle of the left tube and the oil-water mixture is poured into the wide-mouth collection container on the right side. The red oil will quickly pass through the MFA@PDMS and flow into the collection container on the lower left side, while the blue deionized water is blocked on the top side of the MFA@PDMS. Finally, the oil-water mixture is successfully separated completely. When the oil-water mixture comes into contact with the MFA@PDMS surface together, its inherent superhydrophobic and super lipophilic properties allow the oil contaminants to penetrate and pass through the material instantly, while the aqueous solution is blocked by its hydrophobic nature. In this study, after each separation, the sample was washed and dried and used for the next oil-water separation.

  1. The reusability of the Modified sponge is only shown for hexane. What about other oils? and the exact amount of decrement of adsorption capacity is not mentioned after eight cycles. 

Dear reviewers, we sincerely thank you for your valuable suggestions and comments. In this study, the modified sponge has the strongest oil-water separation ability for hexane, the adsorption capacity and oil-water separation capacity of modified sponge for hexane are the main focus in this study. Therefore, we have done tests for typical hexane on top of reusability. After 8 reuse experiments, its oil absorption capacity remained stable and did not decrease significantly.

Finally, we sincerely hope that this revised manuscript has addressed all your comments and suggestions. We appreciated for reviewer’s warm work earnestly, and hope that the correction will meet with approval. Once again, thank you very much for your comments and suggestions. We would like to thank the referee again for taking the time to review our manuscript.

Round 2

Reviewer 1 Report (Previous Reviewer 1)

This manuscript was revised according to the comments, and the quality is much improved. I'm glad to recommend publication.

Reviewer 2 Report (New Reviewer)

I think the manuscript can be considered for publishing.

This manuscript is a resubmission of an earlier submission. The following is a list of the peer review reports and author responses from that submission.

Round 1

Reviewer 1 Report

Title: Super-hydrophobic magnetic fly ash coated polydimethylsiloxane (MFA@PDMS) sponge as an absorbent for rapid and efficient oil/water separation

The authors have made some corrections, but some problems still need to be revised.

General comments: 

1. There are some logical errors that need to be further revised, such as the sentence in abstract “MFA played a vital role to facilitate the uniform MFA particles coated on the skeletons of PDMS sponge”. How does MFA facilitate itself coating on PDMS sponge?

2. The role of fly ash is not clear. Why did the authors use fly ash? Does it endow a special property to oil/water separation? The meaning of using fly ash should make a clarity.

3. The main product is MFA@PDMS, and only TEM of Fe3O4 is meaningless. TEM analysis of MFA@PDMS and Scanning Electron Microscopy analysis of FA, PDMS, Fe3O4 and MFA@PDMS should be provided to make a further morphology comparison of those materials.

4. The FT-IR analysis of FA, PDMS and Fe3O4 should be provided to compare the absorption peaks and to demonstrate that the absorption peaks are caused by the polymerization of fly ash aluminosilicate or silicon-oxygen tetrahedron structure.

5. A lot of careless writing should be revised, for example, the chemical information of FeCl2.4H2O, FeCl3.6H2O, ammonia water, sodium citrate solution and so on should be provided and added into 2.1. Materials. Line 125, the amount of deionized water and n-hexane need to be specified. Besides, what is RHA@PDMS? It should be defined at the first time it is used.

6. Figure 5, the conclusion “it can be observed that the red normal hexane is quickly absorbed by the sponge, the oil-bearing wat body quickly recovers to be clear and transparent,” can not be obtained due to the unclear picture. More clear pictures should be provided.

7. Figure 3 and Figure 7, what effect does this magnetism have on oil-water separation?

8. Line 165-170, references should be provided to support this conclusion.

Author Response

Comments from reviewer 1

  1. There are some logical errors that need to be further revised, such as the sentence in abstract “MFA played a vital role to facilitate the uniform MFA particles coated on the skeletons of PDMS sponge”. How does MFA facilitate itself coating on PDMS sponge?

  Dear Reviewer, thank you for your kind attention to pointing out our mistakes, and we have made corrections to the problems for your review.

Abstract: In this study, magnetic fly ash was prepared with fly ash and nano-magnetic Fe3O4 obtained by co-precipitation method; Then the magnetic fly ash/ polydimethylsiloxane (MFA@PDMS) sponge was prepared via simple dip-coating PDMS containing ethanol in magnetic fly ash aqueous suspension, then solidifying, where Fe3O4 played a vital role to facilitate the uniform FA particles coated on the skeletons of PDMS sponge. After being covered with the PDMS matrix, PDMS contributing super-hydrophobicity to the sponge with a large lubricating oil absorption, and it takes only 10 min for the material to adsorb 6 times its own weight of n-hexane (oil phase). Moreover, the MFA@PDMS sponge possesses outstanding recyclability and stability since no decline in absorption efficiency was observed after more than 8 cycles,and the stress-strain curves of 20 times cyclic compression for the sponge have good overlap, and the maximum stress is basically unchanged, and can be restored to the original shape, indicating that the sponge has good mechanical properties, elasticity, and fatigue resistance.

  1. The role of fly ash is not clear. Why did the authors use fly ash? Does it endow a special property to oil/water separation? The meaning of using fly ash should make a clarity.

  Thank you very much for your suggestions, we have revised the shortcomings, your comments are very valuable to us, please look over them.

  Fly ash is a kind of solid waste, whose main components are silica and alumina, and fly ash itself has adsorption capacity. Compared with other materials, fly ash has a large specific surface area and a porous structure, and thus has good adsorption performance. The use of fly ash as raw material has the advantages of simple process, resource saving and environmental protection.

  For example, Yu et al. firstly prepared Fe3O4 nanoparticles by the solvothermal method, and then used ammonium persulfate (APS) as initiator to synthesize magnetic polystyrene oil-absorbing materials with different coating rates by emulsion polymerization of styrene and divinylbenzene (DVB) at different doses on the nanoparticle surfaces [18].

  1. The main product is MFA@PDMS, and only TEM of Fe3O4is meaningless. TEM analysis of MFA@PDMS and Scanning Electron Microscopy analysis of FA, PDMS, Fe3O4 and MFA@PDMS should be provided to make a further morphology comparison of those materials.

  Dear Reviewer, your comments are very insightful. As a Communication, the main purpose of our study is to highlight the oil-water separation ability of sponge materials, so we want to support this study with short and concise data and information.

  1. The FT-IR analysis of FA, PDMS and Fe3Oshould be provided to compare the absorption peaks and to demonstrate that the absorption peaks are caused by the polymerization of fly ash aluminosilicate or silicon-oxygen tetrahedron structure.

  Dear Reviewer, your comments are very insightful. As a Communication, the main purpose of our study is to highlight the oil-water separation ability of sponge materials, so we want to support this study with short and concise data and information.

  1. A lot of careless writing should be revised, for example, 

  Dear Reviewers, Thank you very much for your valuable comments. We have revised and corrected the deficiencies and ask for your review.

the chemical information of FeCl2.4H2O, FeCl3.6H2O, ammonia water, sodium citrate solution and so on should be provided and added into “2.1. Materials.” 

  1. Materials and Methods

2.1. Materials

  The fly ash (FA) used in this study was obtained from Kanas Power Plant in Xinjiang Autonomous Region, China; It has a chemical composition of SiO2 (45.9 wt%), Al2O3 (19.03 wt%), CaO (16.39 wt%), SO3 (6.01 wt%), Fe2O3 (5.65 wt%), K2O (2.07 wt%), MgO (2.02 wt%), and Na2O (0.954 wt%); PDMS prepolymer and its curing agent are from American Dow Corning; FeCl2·4H2O, FeCl3·6H2O and sodium citrate all come from Shanghai Macklin Biochemical Co., Ltd; Ammonia water was from Sigma-Aldrich (Shanghai) Trading Co., Ltd.

Line 125, the amount of deionized water and n-hexane need to be specified. 

3.2. Hydrophobicity of MFA@PDMS sponge

  Deionized water dyed with blue ink (aqueous phase) and hexane dyed with oil-soluble black (oil phase) were dropped on the surface of MFA@PDMS sponge (the volume ratio of hexane to water was 1:1), and the experimental results are shown in Figure 4. The deionized water with blue ink shows a typical spherical shape on the surface of the material, while the black hexane penetrates into the interior of the material. The material with hydrophobic function cannot be wetted by water, but can be wetted by oil with good oil-water selectivity. It can be seen that the sponge material has good hydrophobicity to aqueous solutions, reflecting the good waterproof performance of the material, which is the key to give the material oil/water separation selectivity.

Besides, what is RHA@PDMS? It should be defined at the first time it is used.

  The particle size of the dispersed phase of the oil-water emulsion is far less than 20 μm, and a stable protective layer is formed on the surface of the droplet molecules, so it is difficult to demulsify the emulsion and make the oil-water separation extremely difficult. The separation efficiency of magnetic MFA@PDMS for different types of emulsified oil (n-hexane, phenol, rapeseed oil) was investigated according to the experimental method of Figure 7, and the results are shown in Figure 8.

  It can be seen from Figure 8 that the separation efficiency of magnetic MFA@PDMS sponge for n-hexane emulsified oil was the highest, reaching 97. 12 %, followed by the adsorption of phenol, reaching 95. 27 %. This is due to the unique structure of fly ash, such as large specific surface area, porosity and abundant aluminum-silicon components, as well as the super-hydrophobic performance of PDMS, which can attract organic molecules and chain-like structures to the inner surface of sponge materials through π-π dispersion interaction and donor receptor effect [18].

  1. Figure 5, the conclusion “it can be observed that the red normal hexane is quickly absorbed by the sponge, the oil-bearing wat body quickly recovers to be clear and transparent,” can not be obtained due to the unclear picture. More clear pictures should be provided.

  Dear reviewer, your comments are very pertinent. We are showing the magnetic properties of the sponge material in this figure. The figure shows an external magnetic field interfering with the sponge material by holding a magnet underneath the Petri dish, and the sponge material moves with the movement of the magnet. We think the picture can demonstrate this property.

  1. Figure 3 and Figure 7, what effect does this magnetism have on oil-water separation?

  Dear Reviewers, Thank you very much for your valuable comments. we have added the role of magnetism to the introduction section and ask for your review.

  The preparation process of magnetic materials is simple, and the magnetic sponge not only has super hydrophobicity and super lipophilicity, but also can achieve the purpose of separating oil-water mixture under the manipulation of the applied magnetic field [17]. Moreover, the magnetic nanomolecular sieve composites have the advantages of high magnetization performance and surface area, and in addition, Fe3O4 nanoparticles can also improve the mechanical properties of the sponge, which remains superhydrophobic when it is strained in tension or compression, and the tested sponge also shows good mechanical stability, oil stability and reusability in terms of superhydrophobicity and oil absorption [15].

  1. Line 165-170, references should be provided to support this conclusion.

  Dear reviewers, we have made changes and clarifications in response to your suggestions, please review them.

  It can be seen from Figure 8 that the separation efficiency of magnetic MFA@PDMS sponge for n-hexane emulsified oil was the highest, reaching 97. 12 %, followed by the adsorption of phenol, reaching 95. 27 %. This is due to the unique structure of fly ash, such as large specific surface area, porosity and abundant aluminum-silicon components, as well as the super-hydrophobic performance of PDMS, which can attract organic molecules and chain-like structures to the inner surface of sponge materials through π-π dispersion interaction and donor receptor effect [18].

  Finally, we sincerely hope that this revised manuscript has addressed all your comments and suggestions. We appreciated for reviewers’ warm work earnestly, and hope that the correction will meet with approval. Once again, thank you very much for your comments and suggestions. We would like to thank the referee again for taking the time to review our manuscript.

Reviewer 2 Report

1.     The “super-hydrophobic” claim did not show the water contact angle (WCA) of MFA@PDMS surface is larger than 150°. In fact, the WAC of general PDMS (Sylgard 184, Dow Corning, USA) with uniform and flat surface is about 109°, which internally provides the hydrophobicity of the PDMS surface. In the meantime, the authors did not mention the reason why the hydrophobicity facilitates the MFA@PDMS in separation.

2.     The authors indicated that the porous MFA@PDMS sponge is a bionic or biomimetic material. However, there is no evidence to show where the inspirations or innovations come from, for instance, any connection between targeted animals, plants, or natural structures. So, it is hard to convince readers that the porous MFA@PDMS is a bionic or biomimetic material. Although the author mentioned some published research about oil/water separation materials inspired from the lotus leaves in introduction, the lack of the demonstration of surface structure or purification properties of MFA@PDMS itself is hard to support the biomimetic idea. Significantly, there is no macro-level pore on lotus leaves’ surface for the published literature.

3.     The TEM image is based on MFA without PDMS as matrix. It is hard to tell that the particle size is uniform with an average diameter ~30 nm. Also, the pure MFA cannot demonstrate the uniformity of MFA after dispersion in PDMS solution, and further in MFA@PDMS material. Usually, the nanoparticles without any surface treatment show strong aggregation in the continuous phase. Also, the single MFA TEM analysis did not show the strong connection with the whole MFA@PDMS sponge oil/water separation study.

4.     There is no detailed description of all characterization tools, including TEM, FTIR, magnetometer, mechanical properties, and oil/water separation test. The characterization procedure and the pre-treatment of each sample are crucial to the consequent results. The author mentioned the X-ray fluorescence spectrometer (XRF) and scanning electron microscopy (SEM) in the characterization part but did not discuss in this paper.

5.     It is hard for readers to understand the statement that the authors claimed in the paper without the previous experimental steps description, especially in section 3.3 part. For example, in line 142, author did not point out how external magnetic field facilitate the separation process; in line 148, the author did not mention the methods used to determine the “absorption time”; in line 151, the author didn’t clearly tell how “reaching 6 time of its own weight” comes from; and in line 153, it is hard to tell why hydrophobicity play an important role in separation.

6.     In section 3.3.2, where does RHA@PDMS come from? How do authors define the separation efficiency? What kind of experiment and characterization did the author conduct by getting these numbers, 97.12%, 95.27% and 91.26%?

7.     The figures, like Figure 3., Figure 4., Figure 5., Figure 7., cannot verify the viewpoints the author described in the paper.

Author Response

Comments from reviewer 2

  1. The “super-hydrophobic” claim did not show the water contact angle (WCA) of MFA@PDMS surface is larger than 150°. In fact, the WAC of general PDMS (Sylgard 184, Dow Corning, USA) with uniform and flat surface is about 109°, which internally provides the hydrophobicity of the PDMS surface. In the meantime, the authors did not mention the reason why the hydrophobicity facilitates the MFA@PDMS in separation.

  Dear Reviewer, Thank you very much for your valuable review comments. We have revised the defective parts you pointed out and added them to the manuscript, please take a look.

  Deionized water dyed with blue ink (aqueous phase) and hexane dyed with oil-soluble black (oil phase) were dropped on the surface of MFA@PDMS sponge (the volume ratio of hexane to water was 1:1), and the experimental results are shown in Figure 4. The deionized water with blue ink shows a typical spherical shape on the surface of the material, while the black hexane penetrates into the interior of the material. The material with hydrophobic function cannot be wetted by water, but can be wetted by oil with good oil-water selectivity. It can be seen that the sponge material has good hydrophobicity to aqueous solutions, reflecting the good waterproof performance of the material, which is the key to give the material oil/water separation selectivity.

  1. The authors indicated that the porous MFA@PDMS sponge is a bionic or biomimetic material. However, there is no evidence to show where the inspirations or innovations come from, for instance, any connection between targeted animals, plants, or natural structures. So, it is hard to convince readers that the porous MFA@PDMS is a bionic or biomimetic material. Although the author mentioned some published research about oil/water separation materials inspired from the lotus leaves in introduction, the lack of the demonstration of surface structure or purification properties of MFA@PDMS itself is hard to support the biomimetic idea. Significantly, there is no macro-level pore on lotus leaves’ surface for the published literature.

  Thank you very much for your careful and patient guidance. We have made changes in the introduction section to address this issue and would appreciate your review.

  With global industrialization, one of the major sources of water pollution, oil spills (organic waste) remain one of the most difficult challenges facing the world today. It is not only harmful to the natural ecosystem, but also has long-term adverse effects on human health and economy [1-2]. However, its treatment methods, such as flotation [3], combustion and linseed oil, have the disadvantages of poor selectivity and low efficiency [4]. In recent years, surface structures of lotus leaves in nature have been mimicked and interfaces similar to hydrophobic biomaterials with completely different wettability for oil and water have been designed and synthesized on the basis of surface chemical bionics, which can effectively avoid these limitations and achieve efficient separation of oil and water. Superhydrophobic materials prepared by mimicking the surface characteristics of natural superhydrophobic materials are collectively known as biomimetic superhydrophobic materials and are rapidly developing in the adsorption of organic pollutants and purification of oily wastewater [5-7].

  1. The TEM image is based on MFA without PDMS as matrix. It is hard to tell that the particle size is uniform with an average diameter ~30 nm. Also, the pure MFA cannot demonstrate the uniformity of MFA after dispersion in PDMS solution, and further in MFA@PDMS material. Usually, the nanoparticles without any surface treatment show strong aggregation in the continuous phase. Also, the single MFA TEM analysis did not show the strong connection with the whole MFA@PDMS sponge oil/water separation study.

  Thank you very much for your insightful comments. By compounding ferric tetroxide with fly ash, MFA can be uniformly dispersed in PDMS, and the magnetic and hydrophobic nature of the material both contribute greatly to oil-water separation.

  1. There is no detailed description of all characterization tools, including TEM, FTIR, magnetometer, mechanical properties, and oil/water separation test. The characterization procedure and the pre-treatment of each sample are crucial to the consequent results. The author mentioned the X-ray fluorescence spectrometer (XRF) and scanning electron microscopy (SEM) in the characterization part but did not discuss in this paper.

  Thank you very much for your suggestion. XRF was used to characterize the oxide composition and content of fly ash and this result is in 2.1; in addition, we have revised the inappropriate part in 2.3, please look it over.

  2.1. Materials

  The fly ash (FA) used in this study was obtained from Kanas Power Plant in Xinjiang Autonomous Region, China; It has a chemical composition of SiO2 (45.9 wt%), Al2O3 (19.03 wt%), CaO (16.39 wt%), SO3 (6.01 wt%), Fe2O3 (5.65 wt%), K2O (2.07 wt%), MgO (2.02 wt%), and Na2O (0.954 wt%); PDMS prepolymer and its curing agent are from American Dow Corning; FeCl2·4H2O, FeCl3·6H2O and sodium citrate all come from Shanghai Macklin Biochemical Co., Ltd; Ammonia water was from Sigma-Aldrich (Shanghai) Trading Co., Ltd.

  2.3. Determination and characterization

  The composite samples were analyzed by an X-ray fluorescence spectrometer (XRF, ARL PERFORM'X, USA) and Fourier transformation infrared spectrometer (FTIR, PerkinElmer, Germany) to characterize the sample properties. The morphologies of samples were observed by transmission electron microscope (TEM, TF30, USA). The magnetic analysis of the samples was conducted by vibrating sample magnetometer (VSM, NINHAO, USA). The stress-strain analysis of the samples was conducted by Electronic universal testing machine (CMT6103, USA).

  1. It is hard for readers to understand the statement that the authors claimed in the paper without the previous experimental steps description, especially in section 3.3 part.

  Thank you very much for your detailed review and valuable comments, we have revised and clarified your comments in the text, please read them.

For example, in line 142, author did not point out how external magnetic field facilitate the separation process;

  The preparation process of magnetic materials is simple, and the magnetic sponge not only has super hydrophobicity and super lipophilicity, but also can achieve the purpose of separating oil-water mixture under the manipulation of the applied magnetic field [17]. Moreover, the magnetic nanomolecular sieve composites have the advantages of high magnetization performance and surface area, and in addition, Fe3O4 nanoparticles can also improve the mechanical properties of the sponge, which remains superhydrophobic when it is strained in tension or compression, and the tested sponge also shows good mechanical stability, oil stability and reusability in terms of superhydrophobicity and oil absorption [15].

in line 148, the author did not mention the methods used to determine the “absorption time”; in line 151, the author didn’t clearly tell how “reaching 6 time of its own weight” comes from;

  2.3. Adsorption capacity, separation efficiency and Reusability test

(1) To evaluate the adsorption capacity of the material, a certain mass of the material was first immersed in the organic solvent and weighed after reaching mass absorption equilibrium. The measurement was repeated three times and the average value was taken. The adsorption capacity Q of the material is calculated according to the following formula (1).

Q = (M1 - M0) / M0 (1)

  Where: Q (g/g) is the mass-based adsorption capacity, M1 (g) is the mass of the sponge after oil absorption, and M0 (g) is the mass of the sponge before oil absorption.

(2) Build oil-water separation device for oil-water separation. Calculate the oil-water separation efficiency R of the sponge according to the formula (2).

R = (Vc) / V0 × 100 % (2)

  Where: R (%) is the oil-water separation efficiency, Vc (mL) is the volume of the original solvent or oil collected after separation, and V0 (mL) is the volume of the original solvent or oil before separation.

(3) The repeatability of the material is initially evaluated by a simple electronic universal testing machine. First, the material is immersed in an organic solvent to achieve maximum absorption and its mass is weighed and recorded as M2. Afterwards, the absorbed oil is recovered by mechanical compression and the mass of the material is weighed again and recorded as M3. The reusability of the material was assessed by absorption-compression cycles, weighing the material to calculate the adsorption multiplier (adsorption multiplier = weight of the material after adsorption of the organic solvent / weight of the material itself).

and in line 153, it is hard to tell why hydrophobicity play an important role in separation.

  Deionized water dyed with blue ink (aqueous phase) and hexane dyed with oil-soluble black (oil phase) were dropped on the surface of MFA@PDMS sponge (the volume ratio of hexane to water was 1:1), and the experimental results are shown in Figure 4. The deionized water with blue ink shows a typical spherical shape on the surface of the material, while the black hexane penetrates into the interior of the material. The material with hydrophobic function cannot be wetted by water, but can be wetted by oil with good oil-water selectivity. It can be seen that the sponge material has good hydrophobicity to aqueous solutions, reflecting the good waterproof performance of the material, which is the key to give the material oil/water separation selectivity.

  1. In section 3.3.2, where does RHA@PDMS come from?

  Thank you very much for your valuable comments, we have corrected the oversight, please take a look.

  The particle size of the dispersed phase of the oil-water emulsion is far less than 20 μm, and a stable protective layer is formed on the surface of the droplet molecules, so it is difficult to demulsify the emulsion and make the oil-water separation extremely difficult. The separation efficiency of magnetic MFA@PDMS for different types of emulsified oil (n-hexane, phenol, rapeseed oil) was investigated according to the experimental method of Figure 7, and the results are shown in Figure 8.

How do authors define the separation efficiency? What kind of experiment and characterization did the author conduct by getting these numbers, 97.12%, 95.27% and 91.26%?

  Dear reviewers, we have added this section and ask you to look through it.

  2.3. Adsorption capacity, separation efficiency and Reusability test

(1) To evaluate the adsorption capacity of the material, a certain mass of the material was first immersed in the organic solvent and weighed after reaching mass absorption equilibrium. The measurement was repeated three times and the average value was taken. The adsorption capacity Q of the material is calculated according to the following formula (1).

Q = (M1 - M0) / M0 (1)

  Where: Q (g/g) is the mass-based adsorption capacity, M1 (g) is the mass of the sponge after oil absorption, and M0 (g) is the mass of the sponge before oil absorption.

(2) Build oil-water separation device for oil-water separation. Calculate the oil-water separation efficiency R of the sponge according to the formula (2).

R = (Vc) / V0 × 100 % (2)

  Where: R (%) is the oil-water separation efficiency, Vc (mL) is the volume of the original solvent or oil collected after separation, and V0 (mL) is the volume of the original solvent or oil before separation.

(3) The repeatability of the material is initially evaluated by a simple electronic universal testing machine. First, the material is immersed in an organic solvent to achieve maximum absorption and its mass is weighed and recorded as M2. Afterwards, the absorbed oil is recovered by mechanical compression and the mass of the material is weighed again and recorded as M3. The reusability of the material was assessed by absorption-compression cycles, weighing the material to calculate the adsorption multiplier (adsorption multiplier = weight of the material after adsorption of the organic solvent / weight of the material itself).

  1. The figures, like Figure 3., Figure 4., Figure 5., Figure 7., cannot verify the viewpoints the author described in the paper.

      Dear reviewers, you have a meticulous insight and spot-on review ability, and we have made significant changes to the entire article. We believe that the logic and results of the entire article now validate our point of view, and would appreciate your review.

Finally, we sincerely hope that this revised manuscript has addressed all your comments and suggestions. We appreciated for reviewers’ warm work earnestly, and hope that the correction will meet with approval. Once again, thank you very much for your comments and suggestions. We would like to thank the referee again for taking the time to review our manuscript.

Round 2

Reviewer 1 Report

There are still many important experiments left out. I understand that it is difficult to supply experiments due to the covid-19, and I am willing to give the author a chance as the author has modified this paper to the maximum extent. Hopefully, after the covid-19, your further research works could publish including those important experiments.

Reviewer 2 Report

1. “Good-hydrophobicity” is not equal to “Super-hydrophobicity” based on the definition. The title of this paper still emphasizes “super-hydrophobic” without demonstration of the surface water contact angle is larger than 150°. For the definition, please see the introduction part of this reference: What do we need for a superhydrophobic surface? A review on the recent progress in the preparation of superhydrophobic surfaces.[1]

2. There is still no specific object, such as any plant, animal, or natural structure to support the bio-inspiration point. The author did not show why the MFA@PDMS is a biomimetic material. In other words, could the author tell the readers what MFA@PDMS mimic?

3. The sample preparation process is crucial, and different methods usually lead to different results. Unfortunately, there is still no detailed description of the characterization part. Authors only listed all tools, but still did not provide the detailed sample preparation procedures or pre-treatment of each characterization.

4. The TEM image of Fe3O4 particles is too weak to demonstrate the uniformity of MFA dispersion in PDMS. TEM analysis of MFA@PDMS or other characterizations should be added.

5. Figure 3, 4, 5, 7 are hard to support the viewpoint in the paper. The author didn’t make any improvements.

[1] What do we need for a superhydrophobic surface? A review on the recent progress in the preparation of superhydrophobic surfaces

DOI: 10.1039/B602486F (Critical Review) Chem. Soc. Rev., 2007, 36, 1350-1368.